# OpenReview forum: "Quantifying Cross-Domain Knowledge Distillation in the Presence of Domain Shift"
_ICML.cc/2026/Conference — ICML 2026 regular_

### Official Review · Reviewer_ke9h · 2026-03-09

**Soundness:** 4
**Presentation:** 3
**Significance:** 2
**Originality:** 3
**Overall Recommendation:** 4
**Confidence:** 3

**Summary:**

This paper provides a high-dimensional asymptotic analysis of cross-domain knowledge distillation. In a linear ridge regression setting, the authors characterize the student model's excess risk under covariate and model shift, providing a bias–variance decomposition. Main results: (1) even under large domain discrepancy, there may exist $\xi$ such that distillation outperforms the student-only baseline; (2) a "crossed double descent" phenomenon: excess risk varies non-monotonically with the dimension-to-sample-size ratios of both teacher and student. The authors also analyze ridgeless regression and discuss the case where $\xi$ can be negative (anti-learning the teacher's supervision).

**Compliance With Llm Reviewing Policy:**

Affirmed.

**Final Justification:**

The author has answered my question very well. I raise my score to 4.

**Key Questions For Authors:**

1. In practical nonlinear/deep KD, what are the applicability bounds of the linear-model conclusions? Is there a plan for simplified deep experiments to validate the trends?
2. Can the authors provide bounds or analysis when $N_1, N_2 \gg M$, i.e., without the $M \sim N_1 \sim N_2$ assumption? How do the main results change in this regime?
3. What is the intuitive explanation for the condition $\xi<0$ in Corollary 3.5 (Eq. 10)? What types of data/tasks are more likely to satisfy it?
4. For the crossed double descent plot (Figure 2), how are the axis ranges for teacher and student dimension ratios chosen? Is there a general rule-of-thumb for the "optimal region"?
5. How does this work's bias–variance decomposition relate to and differ from Menon et al. (2021)'s statistical perspective on distillation?

**Limitations:**

The authors note the analysis is limited to linear models and high-dimensional asymptotics. It would help to briefly discuss in the main text: potential extensions to kernel regression or two-layer networks; connections to W2S generalization and LLM distillation; relaxation of the $M \sim N_1 \sim N_2$ regime (e.g., when $N_1, N_2 \gg M$); comparison with Menon et al. (2021).

**Strengths And Weaknesses:**

**Strengths:**
1. The derivation is rigorous, based on random matrix theory; Theorems 3.1 and 3.3 and Corollaries 3.4–3.5 provide a complete bias–variance characterization of excess risk.
2. The result that distillation can outperform student-only even under large domain discrepancy, and that $\xi$ can be negative (anti-learning the teacher), offers new insights.
3. The discovery of crossed double descent enriches the double descent literature; relaxing the joint diagonalizability assumption generalizes prior work.
4. The structure is clear and notation is well defined; comparison to Hastie et al., Emrullah Ildiz et al., Moniri & Hassani is adequate.
5. Theoretical understanding of cross-domain KD informs model selection and $\xi$ tuning in practice.

**Weaknesses:**
1. The analysis is limited to linear ridge regression; the applicability to practical nonlinear/deep KD (e.g., neural networks, LLMs) is unclear, and there are no experiments—even simplified deep experiments—to validate the trends.
2. The high-dimensional asymptotic regime ($M \sim N_1 \sim N_2$) is rather strong. In practice, it often holds $N_1, N_2 \gg M$. It would strengthen the paper to provide bounds or analysis without this assumption.
3. Some formulas (e.g., Theorem 3.1) are complex. The paper would benefit from more intuitive explanations or numerical examples illustrating the main phenomena.
4. Potential extensions to kernel regression, two-layer networks, or connections to W2S generalization and LLM distillation are not discussed.
5. Menon et al. (2021) also provide a bias–variance decomposition for distillation; a more thorough comparison with their statistical perspective would clarify the distinction and contribution of this work.


Menon, Aditya K., et al. "A statistical perspective on distillation." International Conference on Machine Learning. PMLR, 2021.

---

> ### Author Rebuttal · Authors · 2026-03-30
>
> Thank you for the feedback. Please find below responses to your concerns.
>
> Responses to
> + Q1. Although our theoretical analysis is based on linear models, we believe it provides valuable insights for practical applications. A key contribution of our theoretical results is that $\xi$ can be selected from the entire real line R, specifically allowing for negative values, which was not mentioned in prior literature. To demonstrate that this insight extends to deep models, we conducted experiments on the CIFAR-10 dataset for multi-class classification task using ResNet-18.
> Moreover, we provide an example demonstrating that crossed double descent also emerges in nonlinear models. Due to space limitations, detailed results and discussions are provided in our responses to Reviewer hjcL (Q1 and Q3) and Reviewer SKoL (Q4).
> + Q2.  Our findings naturally extend to the regime where $N_1,N_2\gg M$. Our current analysis utilizes the proportional asymptotic limit ($M\sim N_1\sim N_2$), which is a mathematically more challenging setting. This is because the assumption that
> $M\sim N_1\sim N_2$  is a typical assumption in random matrix theory, which is harder to analysis. When $N_1,N_2\gg M$, the dimension of $\beta$ is much smaller than sample sizes, which is more close to the low-dimensional regime. Specifically, in this regime, $ {X}_i {X}_i^\top/N_i$ becomes a consistent estimator of the population covariance $ {\Sigma}_i$. Consequently, we have $ {Q}_i\asymp  {\Pi}_i=( {\Sigma}_i-z_i {I})^{-1}$ and $ {Q}_i {AQ}_i\asymp  {\Pi}_i {A\Pi}_i $. Thus, one only needs to substitute these terms into the existing results.
> It is worth noting that since $N_1,N_2\gg M$, the variance term converges to zero. This can also be directly observed from Eq.8: since the trace term is of order $O(M)=o(N_i),i=1,2$, this term becomes negligible.
> + Q3. Eq.10 characterizes a regime in which the optimal $\xi$ becomes negative, a phenomenon that carries significant practical implications. Intuitively, this condition is satisfied when the direct supervision from the target domain is sufficiently reliable that the teacher's guidance becomes a hindrance rather than a benefit. For example, when the SNR is sufficiently high to satisfy Eq.10,  the target data are relatively clean, with noise being negligible compared to the signal. In this case, the teacher's predictions may introduce bias, making them less reliable than the ground-truth labels. As a result, assigning a negative weight to the teacher can help counteract this bias. A similar phenomenon arises when $M/N_2$ is sufficiently small. This corresponds to a regime with abundant target samples relative to the model dimension, where the student can learn an accurate model directly from the target data. In such settings, reliance on the teacher becomes unnecessary, again leading to a negative optimal $\xi$.
> + Q4. The theoretical results in Eq.29–32 of the Appendix show that the excess risk diverges to infinity whenever either ratio approaches 1. Therefore, we selected axis ranges that encompass this critical value to clearly visualize this phenomenon.
> The optimal region for the ratios $\gamma_1$
>   and $\gamma_2$ corresponds to the case where both approach zero simultaneously, indicating that the sample size is significantly larger than the dimension. As Eq.29 shows that regardless of the value of $\xi$, $ER\to 0$ as long as $\gamma_1,\gamma_2\to 0$. In contrast, $ER$ converges to a positive limit. This implies that sufficient data leads to superior training performance.
> + Q5. Thank you for highlighting the work by Menon et al. (2021), which has been cited in our manuscript, and which elegantly demonstrates how a teacher reduces the variance of the student’s objective function and provides a criterion for a 'good' teacher. We summarize the key differences below:
>     1. *Generalization to Cross-Domain KD:* While Menon et al. (2021) primarily assume the teacher and student share the same training data, our framework is specifically designed for cross-domain knowledge distillation. We characterize the excess risk under domain shift, allowing our theory to encompass scenarios with substantial discrepancies between the source and target distributions.
>     2. *Discovery of the "Crossed Double Descent":* In addition to the standard bias-variance trade-off discussed in Menon et al. (2021) and prior literature, we identify a novel "crossed double descent" phenomenon.
>     3. *Analysis of the Parameter $\xi$:* Their work considers a setting where the student is trained purely on the soft label (corresponding to the special case $\xi=1$ in our framework). We further discuss the role of $\xi$ and provide a more complete characterization of when and how distillation helps under domain shift.
>
> Responses to
> + Weakness 3&4: We will revise the manuscript to improve readability by adding further explanations of the theoretical results and we will include a discussion of potential extensions.

---

> > ### Author Rebuttal · Reviewer_ke9h · 2026-04-01
> >
> > Thank you for your reply. I will raise the score to 4 points. I cannot give a higher score because the model has not surpassed the linear model.

---

> > > ### Author Response · Authors · 2026-04-02
> > >
> > > Thank you very much for reading our rebuttal and adjusting our score.

---

### Official Review · Reviewer_tQPQ · 2026-03-13

**Soundness:** 3
**Presentation:** 3
**Significance:** 2
**Originality:** 2
**Overall Recommendation:** 4
**Confidence:** 4

**Summary:**

This paper studies cross-domain knowledge distillation using high-dimensional techniques. They give a bias-variance trade-off result. Furthermore, they identify a crossed double descent phenomenon. I think the key idea is in the equation
$L(\xi)=\xi l (y_2^t, y_2^s)+(1-\xi) l(y_2, y_2^s).$
You propose a new optimization task. You think it can help improve cross-domain knowledge distillation.

**Compliance With Llm Reviewing Policy:**

Affirmed.

**Final Justification:**

I raise the score to 4 points. I wish you can add more discussion on \xi and multiple sources in the camera-ready version because of the discussion

**Key Questions For Authors:**

1. For nonlinear cases in practice, what can we still learn from the linear analysis? Does it provide an approximation, a robustness benchmark, or some qualitative guidance?
2. How should $\xi$ be chosen in practice?
3. Is it identifiable from the model assumptions, and can it be consistently estimated from real data?
4. Is there a theoretical or practical reason to require $\xi \in [0,1]$?
5. What changes if $\xi$ lies outside this interval?
6. Can the framework be extended to multi-source settings? For example, can it handle multiple teacher models rather than only two domains?

**Limitations:**

1. The framework relies on a strong linearity assumption.
While this assumption may be justified for analytical tractability, its applicability to real-world settings is less clear, especially when the true data-generating process is nonlinear.

2. The current analysis is restricted to the two-domain setting.
This raises concerns about scalability and generalizability to more realistic multi-source settings, such as cases with multiple teacher models.

**Strengths And Weaknesses:**

Strengths: Their theory is excellent. Using the high-dimensional results, they provide a good bias-variance trade-off that helps us understand knowledge distillation.

Weaknesses and questions:
1. The practical value of the linear assumption is not fully clear.
Even if the linear case is theoretically convenient, it would be helpful to understand what insights, guarantees, or approximations this analysis can provide for nonlinear settings encountered in practice.
2. The role of $\xi$ seems underexplained.
Since $\xi$ appears to be a key hyperparameter in equation (1), the paper would benefit from a clearer discussion of its interpretation, identifiability, and practical importance.
3. The admissible range of $\xi$ is not well motivated.
It is unclear whether $\xi$ must lie in $[0,1]$, and what would happen if it takes values far outside this range, such as $\xi=-100$. The paper should clarify whether restricting $\xi$ to $[0,1]$ brings special advantages, for example through a convex-combination interpretation.

---

> ### Author Rebuttal · Authors · 2026-03-30
>
> Thank you for your valuable comments. Please find our responses to your questions below.
>
> Responses to
> + Q1: While our analysis is linear, we believe it provides valuable guidance for practical KD. Our results reveal that $\xi$ can be selected from the entire real line, including negative values, which appears to have received little attention in previous literature. To demonstrate that this insight extends to deep models, we conducted experiments on the CIFAR-10 dataset for multi-class classification task. Due to space limitations, please refer to our first response to Reviewer hjcL, which contains both the detailed explanation and the experimental results.
>
> + Q2&Q3: We propose a practical estimation method to determine $\xi$ directly from data by leveraging a small validation set. Suppose the sample sizes of the source domain and target domain are $N\_1$ and $N\_2$, respectively.
>  We randomly select a small subset of the target domain of size $n\_2$ as a validation set, where $n\_2\to \infty$
>  and $n\_2=o(N\_2)$. We denote this validation set by $(x\_i,y\_i)\_{i=1}^{n\_2}. $ Let $f\_s^\xi$
>  be the student model corresponding to the imitation parameter $\xi$, and denote its predictions on the validation set by $\hat{y}\_{\xi,i}=f^\xi (x\_i).$ We estimate the optimal $\xi$ by solving
>
>      $$\xi^*=\arg\min\_\xi \frac{1}{n\_2}\sum\_\xi|\hat{y}\_{\xi,i}-y\_i|^2.$$
>
>     Under the assumption that  target domain samples are i.i.d., the law of large numbers ensures that $\xi^*$ is a estimator of the optimal $\xi$. Due to space limitations, please refer to our reply to SKoL's Q1 for the numerical verification of this estimation method.
> + Q4&Q5: We wish to clarify that our framework does not require $\xi\in [0,1]$, despite this being a common restriction in much of the existing theoretical and empirical literature. On the contrary, our theoretical analysis reveals that the optimal $\xi$ can take values across the entire real line, as demonstrated in Proposition 3.9. This phenomenon arises because, under our setting with other parameters fixed, the asymptotic excess risk is a quadratic function of $\xi$.
> While Das \& Sanghavi (2023) theoretically proved that the optimal $\xi$ can exceed 1, our work further extends their findings by providing theoretical instances where the optimal $\xi$ is negative (see Corollary 3.5).
> Furthermore, in response to your Question 1, we provide an additional example using deep models to empirically validate these theoretical insights.
>
> + Q6: Yes. Extending our framework to multi-source settings involving multiple teacher models represents a natural generalization of our work. We now present the two-teacher case (from different domains) for illustrative simplicity. Consider two teacher models with parameters $\beta_{t0}$ and $\beta_{t1}$, trained on the datasets $(X_0,y_0)$ and $(X_1,y_1)$, respectively. Their predictions on the target domain data $X_2$ are denoted as $y_{t0} = X_2^\top \beta_{t0}$ and $y_{t1} = X_2^\top \beta_{t1}$.
> The extended loss function  is:
> $$
> l(\beta) = \frac{\xi_1}{N_2} \\| y_{t0} - X_2^\top \beta \\|^2 + \frac{\xi_2}{N_2} \\| y_{t1} - X_2^\top \beta \\|^2 + \frac{1 - \xi_1 - \xi_2}{N_2} \\| y_{2} - X_2^\top \beta \\|^2 + \lambda_s \\| \beta \\|^2.
> $$
> The student model becomes
> $$
> \begin{aligned}
> \beta_s =(X_2 X_2^\top + N_2 \lambda_s I_M)^{-1}[\xi_1 X_2X_2^\top\beta_{t0}+\xi_2X_2X_2^\top\beta_{t1}+(1-\xi_1-\xi_2)X_2(X_2^\top\beta+\varepsilon_2)].
> \end{aligned}
> $$
> Denote the true parameter vector of target domain by $\beta$.
> We need to derive the limit of $(\beta_s-\beta)^\top \Sigma_2(\beta_s-\beta).$
> The multi-teacher model yields cross-terms such as
> $$
> \xi_1\xi_2(Q_2X_2X_2^\top\beta_{t0}/N_2-\beta)^\top\Sigma_2(Q_2X_2X_2^\top\beta_{t1}/N_2-\beta ).
> $$
> Substituting $\beta_{ti}=\frac{1}{N_i}Q_iX_i(X_i^\top\beta_i+\varepsilon_i),i=0,1,$ and omitting constants, this term comprises products of: $$\beta,\beta_0,\beta_1,\Sigma_2,Q_2,Q_2\Sigma_2Q_2, Q_0,Q_1,Q_0X_0\varepsilon_0, Q_1X_1\varepsilon_1,$$
> where $\beta_0$ and $\beta_1$ are parameter vectors of the respective teacher models.
> Under the assumption of domain independence, the limiting behavior of these terms can be established by applying the same arguments presented in the Appendix. For example,
>     $$
> \xi_0\xi_1\frac{1}{N_2N_0N_1}\beta_{0}^\top X_0X_0^\top Q_0X_2X_2^\top Q_2\Sigma_2Q_2X_2X_2^\top Q_1X_1X_1^\top \beta_{1}
> $$
>     $$
> = \xi_1\xi_2\beta_0^\top(I+z_0Q_0) (I+z_2Q_2)\Sigma_2(I+z_2Q_2)(I+z_1Q_1)\beta_1
> $$
>     $$
> \to \xi_0\xi_1 \beta_0^\top (I+z_0\Pi_0)(\Sigma_2+2z_2\Pi_2\Sigma_2+z_2^2\Pi_2\mathcal{S}_2(\Sigma_2)\Pi_2) (I+z_1\Pi_1)\beta_1.
> $$
>     In scenarios involving more teachers, the derivation remains analogous, requiring only the inclusion of additional cross-terms with similar functional forms.

---

> > ### Author Rebuttal · Reviewer_tQPQ · 2026-04-04
> >
> > Thank you for your reply. I will raise the score to 4 points.
> > I wish you can add more discuss on xi and multiple source in camera-ready version

---

> > > ### Author Response · Authors · 2026-04-04
> > >
> > > Thank you for your reply and for raising the score. We will add the discussion on xi and multiple sources as suggested.

---

### Official Review · Reviewer_SKoL · 2026-03-13

**Soundness:** 3
**Presentation:** 2
**Significance:** 3
**Originality:** 3
**Overall Recommendation:** 4
**Confidence:** 2

**Summary:**

The paper provides a high-dimensional theoretical analysis of cross-domain knowledge distillation (KD), where a teacher trained on source domain data provides pseudo-labels for training a student on target domain data with domain shift. Specifically, the paper characterizes the excess risk while accounting for both model and covariate shifts. The results theoretically demonstrate that even when the source and target domains differ substantially, there still may exist a regime where the student model achieves superior generalization ability over the student-only baseline. Furthermore, the paper also identifies a crossed double descent phenomenon, i.e., when the student is trained on a mix of teacher supervision and real labels, the student can inherit two different interpolation instabilities from both teacher estimation and student estimation, hence the excess risk can vary non-monotonically with the teacher's and student's dimension-to-sample-size ratios.

**Compliance With Llm Reviewing Policy:**

Affirmed.

**Final Justification:**

The rebuttal has addressed my clarification questions, and I think the analysis provides meaningful insight. However, since the analysis is limited to linear regression, I am still not confident enough to raise my score to accept. Therefore, I keep my score unchanged at weak accept.

**Key Questions For Authors:**

In Remark 3.10: "We provide further details in Appendix B.8, where we provide closed-form expressions for the optimal $\xi^\*$ under several common settings and demonstrate that covariate shift can, in some cases, be beneficial for KD." Could the authors provide simulation for that, i.e., estimate $\xi^\*$ and show $\xi^\*$ achieves best?

Any guidance for choosing $\xi$ in practice?

Section 5.1 shows neural network experiments. How do you interpret the theory's predictions for networks with feature learning (not just kernel regime)?

In KD, teachers are usually considered as strong. In that case, does the crossed double descent phenomenon still provide practical insight?

**Limitations:**

The theory applies to ridge regression

**Strengths And Weaknesses:**

**Strengths:**

The paper provides theoretical analysis of an important problem in practice where a teacher trained on source domain is used for training a student on target domain data with domain shift. It proves KD can beat student-only training even under significant domain discrepancy.

Clean setup: teacher ridge on source, student ridge on target with KD mixture, and evaluates excess risk on the target distribution, which provides a rigorous theoretical framework for analyzing cross-domain generalization.

Deriving explicit bias-variance decomposition. Corollary 3.5 showing when optimal $\xi < 0$ is genuinely insightful, explaining "anti-learning" phenomena.

Numerical Simulations closely match theoretical analyses, validating the asymptotic formulas.

&nbsp;

**Weaknesses:**

The paper presentation is mathematically heavy; more interpretation could help. I acknowledge that space is limited for dense theory papers. Maybe smaller figures and adding more explanations or providing a short version of theorems with intuition discussion (leaving formal theorems in appendix) may help.

Assumption 2.2 may be violated in practice, e.g., high-dimensional data often lies near a low-dimensional manifold embedded in the high-dimensional space and embeddings from deep neural networks tend to exhibit high anisotropy.

---

> ### Author Rebuttal · Authors · 2026-03-30
>
> Thank you for your valuable comments. Please find our responses below
>
> Responses to
> + Q1. We now provide a method to estimate the optimal $\xi$.  We randomly select a small subset of the target domain of size $n_2$ as a validation set, where $n_2\to\infty$ and $n_2=o(N_2)$. We denote this validation set by $(x_i,y_i)\_{i=1}^{n_2}$. Let $\beta_s^\xi$
>  be the student model corresponding to the imitation parameter, and denote its predictions on the validation set by $\hat{y}\_{\xi,i}=x\_i^\top \beta_s^\xi.$ We estimate the optimal $\xi$ by solving
>
>     $$\xi^\*=\arg\min\_\xi \frac{1}{n\_2}\sum\_i|\hat{y}\_{\xi,i}-y\_i |^2 .$$
>
>    Since the limit of the risk is a quadratic function of $\xi$, by the law of large number, $\xi^*$ is a consistent estimator of the optimal $\xi$.
>     We now perform simulations to verify the effectiveness of this method under a covariate shift between the source and target domains. Specifically, the covariance matrices are defined as $\Sigma_1=I$ and $\Sigma_2=0.5^{|i-j|} $. The coefficient vector $\beta$ is sampled with entries drawn i.i.d. from  the distribution $P(X=1)=P(X=-1)=0.5$, and $\beta$ is shared by both the teacher and student. We set the dimension to $M=200$ and the sample sizes to $N_1=N_2=500$. The input data are i.i.d. random vectors following $N(0, I_M)$.
>
>     Figure 1 illustrates the estimate of $\xi$ obtained via our proposed method using grid search. Figure 2 depicts the average risk, estimated using a large-scale validation set generated independently for each $\xi$; this serves as an empirical approximation of the theoretical risk. The optimal $\xi^*$ estimated by our method is 0.55, which is close to the theoretical value of 0.51.
> + Q2. We believe the strategy outlined in the first reply extends naturally to KD with more complex architectures. In practice, we recommend selecting a satisfactory $\xi$ by leveraging a small validation set, which yields robust performance without the need for theoretical derivation.
> + Q3. While our analysis is linear, we believe it provides valuable qualitative guidance for practical KD. Our results reveal that
>  can be selected from the entire real line, including negative values $\xi$, which appears to have received little attention in previous literature. To demonstrate that this insight extends to deep models, we conducted experiments on the CIFAR-10 dataset for multi-class classification task.  Due to space limitations, please refer to our response to Reviewer hjcL's Q1.
>
> + Q4. We want to clarify that the crossed double descent phenomenon also appears in nonlinear architectures when teacher is stronger. To illustrate this, we conduct experiments under a random feature model. Specifically, both the teacher and the student are trained on data generated  from the same underlying linear relationship:
>     $$
> y=\beta^\top x+e,
> $$
> where $\beta\sim N(0,I_d),x\sim N(0,I_d)$ and $e\sim N(0,0.25).$
> The teacher and student are trained on independent datasets with sample sizes $n_1$ and $n_2$ , respectively. We model both the teacher ($f_1$) and the student ($f_2$) as two-layer MLPs:
> $$
> f_i(x)=w_i^\top \sigma(W_ix), \; i=1,2,
> $$
> where $\sigma$ is tanh, $W_i\in R^{p_i\times d} $ is a random weight matrix whose entries are i.i.d. normal random variables with variance $1/d$.
> Distinct from the linear setting, we define the dimension-to-sample ratios  as $p/n_1$ and $p/n_2$, rather than $d/n_1$ and $d/n_2.$ We fix $d=50,p_1=200, p_2=100$ and choose different $n_1$ and $n_2$ in the experiment.
> Since $p_2>p_1$, the teacher is considered stronger.
> We optimize $w_1$ and $w_2$
>  separately for the teacher and student models, respectively, with a regularization coefficient of $10^{-5}$ and $\xi=0.5.$
>  The results are shown in Figure 3, where the reported risk is averaged over 10 independent runs. As illustrated, the crossed double descent phenomenon persists in this nonlinear setting. This observation highlights an important practical implication: the choice of teacher model should be guided by the sample size of the source domain,  rather than simply opting for the largest architecture; doing so helps avoid falling into the region where the risk can diverge.
>
> Responses to
> + Weakness 1: We will revise the manuscript to improve its readability by providing more intuitive interpretations and explanations. Figure sizes will also be adjusted as needed.
> + Weakness 2: We would like to clarify that Assumption 2.2 already accommodates the low-dimensional embedding scenario to some extent. Specifically, it only requires that a subset of the eigenvalues are non-zero. Moreover, the assumption allows high anisotropy in the input data, as it imposes no constraints on the eigenvectors.
>
> 1. Figure 1 available at: https://anonymous.4open.science/r/ICML26-9682-FIG-6E04/vad.png
> 2. Figure 2 available at: https://anonymous.4open.science/r/ICML26-9682-FIG-6E04/vad1.png
> 3. Figure 3 available at: https://anonymous.4open.science/r/ICML26-9682-FIG-6E04/rf1.png

---

> > ### Author Rebuttal · Reviewer_SKoL · 2026-04-03
> >
> > I thank the authors for their thorough response. The rebuttal has addressed my questions, and I have no further questions.

---

> > > ### Author Response · Authors · 2026-04-03
> > >
> > > We sincerely appreciate your thoughtful feedback. We will revise the manuscript accordingly.

---

### Official Review · Reviewer_hjcL · 2026-03-14

**Soundness:** 3
**Presentation:** 2
**Significance:** 3
**Originality:** 2
**Overall Recommendation:** 4
**Confidence:** 3

**Summary:**

This paper studies when cross-domain knowledge distillation can improve generalization under domain shift. Motivated by the observation that teachers trained on different domains may still provide useful signals, the authors analyze a teacher–student framework where the student learns from both target labels and teacher predictions. They propose a formulation controlled by an imitation parameter and derive theoretical risk expressions using bias–variance analysis in high-dimensional linear models. The theory predicts conditions where cross-domain KD helps and reveals a crossed double descent phenomenon. Experiments and simulations support these theoretical findings. Overall, the paper provides a clear theoretical perspective on when cross-domain KD may be beneficial.

**Compliance With Llm Reviewing Policy:**

Affirmed.

**Key Questions For Authors:**

1. To what extent do the main theoretical results depend on the linear teacher–student assumption?
Could the authors clarify which parts of the theory are expected to generalize to nonlinear KD settings?

2. Why does the paper not include experiments on realistic deep knowledge distillation benchmarks?
Such experiments might help demonstrate whether the theoretical insights translate into practical benefits.

3. The paper identifies a crossed double descent phenomenon.
Could the authors further explain how this observation may guide practical choices of teacher or student capacity?

**Limitations:**

yes

**Strengths And Weaknesses:**

strength
1. The paper focuses on a clear theoretical question: when cross-domain knowledge distillation helps under domain shift. Rather than proposing another empirical algorithm, it analyzes the structure of excess risk under a stylized but interpretable model. This focus gives the paper a well-defined research objective.

2. The theoretical analysis is relatively comprehensive. The paper studies ridge regression, ridgeless regression, and both deterministic and random parameter settings. The derivations are organized around a bias–variance decomposition, which makes the theoretical narrative coherent.

3. The work produces several nontrivial theoretical insights. In particular, it shows that cross-domain KD can outperform a student trained only on target labels even when domain discrepancy exists. It also identifies a crossed double descent phenomenon that extends classical double descent behavior.

weakness
1. The theoretical assumptions are relatively strong. The analysis relies on high-dimensional linear teacher–student models and structured covariance assumptions. While these settings allow precise analysis, they are still far from the complexity of modern deep knowledge distillation.

2. The empirical validation is limited. Most experiments are numerical simulations designed to verify the theoretical predictions rather than evaluations on real datasets. This makes it harder to assess the practical relevance of the theoretical insights.

3. The nonlinear extension remains relatively shallow. The nonlinear experiment in Section 5 explores a simplified neural network setting but does not provide theoretical guarantees comparable to those developed for linear models.

---

> ### Author Rebuttal · Authors · 2026-03-30
>
> Thank you for your valuable comments. We hope our responses below address your specific concerns.
>
> Responses to
> + Question 1: While our analysis is linear, we believe it provides valuable guidance for practical KD. Our  results reveal that $\xi$ can be selected from the entire real line, including negative values, which appears to have received little attention in previous literature. To demonstrate that this insight extends to deep models, we conducted experiments on the CIFAR-10 dataset for multi-class classification task. We employ a ResNet-18 architecture pre-trained on ImageNet. Following the protocol of linear probing, we froze the pre-trained backbone and trained only a softmax classification head. The teacher model was trained on the original RGB images of CIFAR-10 (source domain). Subsequently, the student model was trained on grayscale images (target domain). The target domain was constructed by converting RGB images to grayscale using the ITU-R 601 luma transformation standard.
> The optimization objective is defined as a weighted combination of the standard cross-entropy loss and the KL divergence between the teacher's and student's output distributions:
> $$L= (1 - \xi)L_{CE} + \xi L_{KL},$$
> where the temperature parameter is fixed at 3.0. Optimization was performed using SGD with a learning rate of 0.1, momentum of 0.9, weight decay of $10^{-4}$, and a batch size of 128 for the student classifier head. The results are reported in the following table:
> | ξ | Mean accuracy (%) | sd |
> |:---:|:---:|:---:|
> | -0.3 | 52.89 | 0.12 |
> | -0.2 | 60.12 | 0.14 |
> | -0.1 | 64.07 | 0.29 |
> | 0.0 | 62.21 | 0.38 |
> | 0.1 | 61.79 | 0.13 |
> | 0.2 | 60.93 | 0.22 |
> | 0.3 | 60.23 | 0.38 |
> | 0.5 | 57.17 | 0.36 |
> | 1.0 | 51.70 | 0.17 |
>
>     Due to domain shift (specifically, covariate shift in our theoretical analysis), the teacher model's predictive distribution exhibits significant bias, yielding "negative knowledge." Our results show that the student model achieves optimal performance at a distillation weight of $\xi=-0.1$. This not only outperforms the baseline training ($\xi=0$), but also significantly mitigates the misleading guidance from the teacher. Furthermore, under the pure distillation setting ( $\xi=1$), the student's performance degrades substantially. These findings empirically validate the necessity of incorporating anti-teacher supervision (negative $\xi$) in cross-domain KD.
>
> + Question 2: Our experimental results empirically demonstrate the benefit of choosing $\xi$ across the entire real line; These results directly support the insight from our theory. Please refer to our response to Question 1 for details.
>
> + Question 3: Our analysis demonstrates that the student's generalization ability depends not only on its capacity but also on the interaction between the dimension-to-sample-size ratios of the student and the teacher.  Specifically, for linear models, this phenomenon reveals regions where excess risk diverges to infinity, indicating that dimension-to-sample-size ratios leading to such divergence should be avoided in practice.
>
>     Notably, the crossed double descent phenomenon also appears in nonlinear architectures. To illustrate this, we conduct experiments under the random feature model. We assume that both the teacher and the student are trained on data generated  from the same underlying linear relationship:
>
>     $$ y=\beta^\top x+e,$$
>     where $\beta\sim N(0,I_d),x\sim N(0,I_d)$ and $e\sim N(0,0.25).$ Specifically, the teacher and student utilize independent training datasets  with sample sizes  $n_1,n_2$, respectively. We let both the teacher (denoted by $f_1$) and the student (denoted by $f_2$)  be two layer MLPs:
>     $$ f_i(x)=w_i^\top \sigma(W_ix), \; i=1,2,$$
>     where $\sigma$ is the tanh activation function, $W_i\in R^{p\times d} $ is a random weight matrix whose entries are i.i.d. normal random variables with variance $1/d$.  Distinct from the linear setting, we define the dimension-to-sample ratios  as $p/n_1$ and $p/n_2$ rather than $d/n_1$ and $d/n_2.$ We fix $d=50,p=400$ and choose different $n_1,n_2$ in this experiment. We optimize $w_1$ and $w_2$  separately for the teacher and student models, respectively, with a regularization coefficient of $10^{-5}$ and $\xi=0.5.$
>     The result is shown in https://anonymous.4open.science/r/ICML26-9682-FIG-6E04/rf.png, where the risk  is the average over 10 independent runs. As illustrated in the figure, the crossed double descent phenomenon persists. This implies that, for given sample sizes of both domains, the dimensions of the top layers (hidden layer widths) for both the student and the teacher must be carefully selected to prevent the risk from becoming too large.
>
> Responses to
> +  Weakness 1&2: Please refer to our responses to Questions above.
>
> + Weakness 3: Extending the theoretical guarantees to more complex architectures is a valuable direction, and we plan to pursue this in future work.

---

### Decision · Program_Chairs · 2026-04-30

**Decision:**

Accept (regular)

**Comment:**

The paper presents a theoretical study of the generalization capabilities of cross-domain generalization. Using high-dimensional analysis, the paper characterizes the excess risk of the student model under covariate shifts. The results show that there exists a regime where the student model achieves superior generalization ability over the student-only baseline. Additionally, paper demonstrated a crossed double descent phenomenon.


The following strengths have been identified in the reviews:
-the paper studies a relevant problem,  it provides a bias-variance trade-off that helps to understand knowledge distillation.
-The theoretical analysis is sound and is non trivial.  it shows that KD can be better than student-early training even under domain shift.
-identification of a double descent phenomenon.
-numerical simulations match theoretical analyses.
-Discovery of crossed double descent enriches the double descent literature

The following weaknesses have also been mentioned:
-Strong theoretical assumptions (high-dimensional linear teacher-student models, structured covariance assumptions, asymptotic regim) that may not very realistic
-Limited empirical evaluation (no evaluation on real datasets), results limited to regression.
-Nonlinear extension relatively limited.
-Paper dense, some formulas are complex, the paper needs more intuitive explanations.
-the role of hyper parameter $\xi$ is not clear and sufficiently explained even though the interpretation as anti-learning the teacher is interesting.

Authors have provided multiple details answers during rebuttal.
After rebuttal,
reviewer SKoL was fully satisfied by the answers and kept a weak accept evaluation,
reviewer tQPQ was partially satisfied but raised his score to weak accept and mentioned he would to see more discussions on multi-source and $\xi$ in the final version.
reviewer  ke9h raised his score to weak accept, he did not go higher due to the limitation of linear model.
reviewer hjcL did not provide acknowledgment but did not react negatively.

Overall, all reviewers gave a positive evaluation for this paper. Even if there exist some weaknesses, the paper provides a sound and novel theoretical results that contribute to better understand knowledge distillation and its capacity of generalization. This is clearly a point that is interesting for a part of the ICML community. I think that the work is nice in its scope.
I propose then acceptance.